# Precision Recruitment and Engagement of Individuals at Risk for Diabetes and Hypertension in Clinical Trials (PREDHICT): A Randomized Trial for an E-Persuasive Mobile Application to Inform Decision Making about Clinical Trials

**DOI:** 10.3390/ijerph20237115

**Published:** 2023-11-27

**Authors:** Azizi Seixas, Shannique Richards, Jesse Q. Moore, Chigozirim Izeogu, Laronda A. Hollimon, Peng Jin, Girardin Jean-Louis

**Affiliations:** 1Department of Informatics and Health Data Science, Miller School of Medicine, University of Miami, Miami, FL 33136, USAlxh799@med.miami.edu (L.A.H.); 2Department of Psychiatry and Behavioral Sciences, Miller School of Medicine, University of Miami, Miami, FL 33136, USA; girardin.jean-louis@miami.edu; 3Clinical Psychology, City College, City University of New York, New York, NY 10031, USA; slr2172@tc.columbia.edu; 4Department of Neurology, University of Texas Health Science Center at Houston, Houston, TX 77030, USA; 5Department of Population Health, NYU Grossmann School of Medicine, New York, NY 10016, USA

**Keywords:** decentralized trials, clinical trial, health literacy, altruism

## Abstract

The primary objective of this randomized trial was to test the effectiveness of the PREDHiCT digital application, which provides educational and supportive navigation to increase willingness to participate in a future clinical trial. The second objective was to test whether PREDHiCT increased clinical trial literacy or enhanced psychological facilitators of clinical trial participation, such as altruism. To test these two objectives, we conducted a 1-month remote decentralized trial with 100 participants who either have a personal or family history of cardiometabolic health conditions, such as hypertension, diabetes, and obesity. Results indicated significant changes in altruism (mean: −2.94 vs. 0.83; *p*-value = 0.011) and clinical trial literacy (mean: 0.55 vs. 2.59; *p*-value = 0.001) from baseline to 1-month follow-up between the control and intervention groups. Additionally, participants exposed to personalized clinical trial navigation had greater clinical trial literacy at the end of the study relative to the individuals in the control arm of the study. Our findings indicate that tailored education, navigation, and access to clinical trials—three unique features of our PREDHiCT app—increased altruism and clinical trial literacy but not willingness to participate in a trial.

## 1. Introduction

The FDA recently revealed that almost half of clinical trials failed to include sufficient representation of racial/ethnic minorities and low income individuals [1,2,3,4,5]. The under-representation of these groups in clinical trials is complicated and involves numerous barriers and underlying causes. The commonly held view attributes this lack of diversity to individual-level barriers, such as lack of awareness about clinical trials, limited access, disinterest, mistrust, fear, reluctance to take risks, ethical beliefs, and the perception that there are no direct benefits to participating [6,7]. Despite some moderately successful individual-, provider- (negative attitudes toward patients and negative perceptions of the health system), community-, and system-level ameliorative solutions, a lack of diversity in clinical trials still persists [2,3,4,5,6,7,8,9,10].

In the ongoing pursuit of addressing the persistent issue of underrepresentation in clinical trials, the role of education and navigation solutions has gained increasing attention. Historically, clinical trials have struggled to recruit a diverse participant pool, leading to concerns about the generalizability of research findings. Education and navigation solutions have emerged as valuable tools for dismantling barriers that historically deter individuals from underrepresented backgrounds. The strength of these solutions lies in their potential to provide accessible information and support, helping individuals overcome challenges such as limited awareness, mistrust, and logistical barriers. However, their weakness lies in a lack of precision and personalization. Standardized education and navigation approaches may not adequately address the unique needs and concerns of each potential participant. Additionally, while these solutions can mitigate certain barriers, they may not fully overcome the geographical and logistical challenges that can make participation difficult. 

### 1.1. The Value of Decentralized Trials

Decentralized trials offer a viable solution to addressing persistent access hurdles to participating in clinical trials. By allowing participants to engage in trials from a place of their choosing (t residences and community), we significantly reduce the burden of travel, thereby enhancing accessibility for individuals who might otherwise be excluded due to distance, travel costs, or other constraints. A decentralized approach holds great promise for removing barriers and fostering greater representation in clinical trials. Decentralized trials can also better accommodate diverse populations’ needs, including geographical constraints, cultural considerations, and socioeconomic factors, ensuring equitable representation and improving health outcomes for all [11,12,13].

In sum, a decentralized clinical trial can address the knowledge, access, and navigation limitations of traditional centralized clinical trials [14,15,16,17,18,19,20]. Decentralized trials have demonstrated value in enhancing participant diversity, recruitment, and retention rates and providing more representative real-world data for improved generalizability of study findings [20]. Decentralized trials have gained traction since the COVID-19 pandemic, where in a national poll 56% of individuals preferred a decentralized and virtual clinical trial experience, in contrast to only 29% who would commute to a clinical trial site [20]. Decentralized trials enhance participant-centricity, accessibility, and user-friendliness [21], and have proved more effective in enrolling participants and increasing their trial experience satisfaction compared to traditional trials [22]. 

### 1.2. The Value of Continuous Engagement during Clinical Trials

Recognizing the ongoing challenge of participant retention in clinical trials, there’s a growing consensus on the critical need for continuous engagement strategies to address historical barriers and improve outcomes for diverse participants. To address these challenges, recent research has emphasized the value of continuous engagement, including personalized health tracking, culturally tailored educational materials, and digital nudges. These strategies align trial participation with daily behaviors, making clinical research more personal, practical, and engaging. A key objective of continuous engagement is to fill knowledge gaps and enhance trial literacy. By providing real-time insights, promoting healthy lifestyle behaviors, and reinforcing the value of clinical trial participation, continuous engagement in clinical trials aim to bolster participant retention and willingness to engage in future trials [23]. In light of the growing interest in decentralized trials and their potential to enhance diversity in trial participation, the role of continuous engagement strategies takes on added significance. Providing real-time insights and nudges to adhere to healthy lifestyle behaviors can increase knowledge and value in participating in clinical trials and may enhance participants’ willingness to partake in future trials [24,25,26]. The combination of personalized education and navigation along with a decentralized trial framework holds promise for creating a more inclusive and equitable clinical trial landscape.

### 1.3. Study Objectives

The study objective was to enhance participants’ willingness, motivation, trial literacy, or altruism towards trial participation through a personalized and decentralized education and navigation app compared to the traditional one-size-fits-all centralized trial experience. This app was designed to inform, educate, engage, support, and navigate participants through clinical trial experience, with a strong focus on personalization. To accomplish this, we first conducted qualitative assessments to identify facilitators for increasing participation in decentralized trials. Then, we incorporated these elements into the app’s design and functionality. Our comprehensive approach aims to assess whether this personalized strategy significantly improves participant adherence, retention rates, and overall engagement throughout the clinical trial process. Furthermore, we aim to explore the potential impact of these strategies on the representation of historically underrepresented groups in clinical trials, offering a wide variety of research volunteer opportunities, including clinical trial participation, to cater to diverse individuals and encourage their engagement in research studies. 

## 2. Materials and Method

### 2.1. Design and Setting

Our study was executed in three distinct phases: Phase 1 entailed a qualitative investigation aimed at delineating barriers and facilitators (see Appendix A) of participating in traditional and decentralized trials. In Phase 2, we created the PREDHiCT mobile application, as a decentralized trial navigator and to support daily health tracking. In Phase 3, we conducted a randomized controlled trial to evaluate whether the dynamic PREDHiCT app, customized for lifestyle management and clinical trial education, could improve knowledge, awareness, and willingness to participate in clinical trials compared to the conventional trial experience characterized by static pamphlet-based educational materials, infrequent communication, and limited personalized support. The study was approved by the New York University School of Medicine Institutional Review Board and adhered to all ethical principles of the Declaration of Helsinki. The study was registered at ClinicalTrial.gov (NCT04286113).

### 2.2. Phase 1

The primary aim of the study was to identify barriers and facilitators impacting clinical trial participation. This involved assessing the potential of a mobile application to mitigate these barriers and enhance participation rates among subjects. Furthermore, the study aimed to discern the optimal messaging and strategies required to motivate subjects to engage in clinical trials, ascertain preferred sources and media for receiving information, and determine pertinent content for the personalized newsfeed feature within the mobile app. Additionally, we created a journey map depicting the path to clinical trial participation, which enabled the identification of bottlenecks, decision points, and associated barriers and facilitators throughout the clinical trial journey.

#### 2.2.1. Qualitative Data Collection

In our qualitative study, we organized a focus group that took place over two sessions (one in Phase 1 and the other in Phase 2), each lasting 90 min, involving the same seven (7) participants (3 men and 4 women). We had open conversations with all seven individuals, led by the principal investigator, using a structured interview guide to understand their smartphone usage, experiences with health apps, familiarity with clinical trials, recommend features needed to design a clinical trial digital app, and factors that influence their willingness to participate in a clinical trial (see Table 1). To better grasp the factors influencing clinical trial involvement, we provided a clinical trial journey map (Figure 1) showing the steps and process of taking part in a clinical trial. This helped participants highlight the things that make it difficult to join a trial (barriers) and suggest ways to make being part of a trial easier (facilitators). The discussions were conducted in English and flowed naturally, lasting 90 min per session. Before we started, everyone gave written consent to take part, and all participants were at risk for cardiometabolic health conditions such as obesity, high blood pressure, and metabolic syndrome.

#### 2.2.2. Qualitative Data Analysis

Focus groups were audio-recorded, transcribed verbatim, and analyzed using summative content analysis by Datagain Analytics. Content analysis is a form of qualitative data analysis that quantifies qualitative data to determine the intensity (frequency) and contextual quality via comparison of certain responses to derive underlying context, interpretations, meanings, and theories of data. During this process, transcripts were read multiple times by three coders and coded line by line into meaningful segments, recurring concepts, and emergent themes. 

### 2.3. Phase 2: Development of PREDHiCT Mobile Application

In Phase II, we conducted a second focus group with the same 7 participants from Phase 1. Specifically, we conducted heuristic, acceptability, and usability testing of the incorporated sections of the PREDHiCT mobile application (see Figure 2. The purpose of Phase II was to receive feedback about important elements that should be included in the app to increase adherence to clinical trial participation. The purpose of the PREDHiCT digital tool is to increase patient awareness and knowledge about clinical trial opportunities, help navigate individuals at risk for cardiometabolic conditions to appropriate clinical trial opportunities, and increase awareness/knowledge, attitude, motivation, willingness to participate, self-efficacy, and information contagion behaviors (sharing information about clinical trials and/or signing up for health-related social network groups) about clinical trials. Focus group participants recommended that we develop a personalized newsfeed with curated content about health conditions that pertain to the participant. To fulfill this recommendation, we developed and trained an artificial intelligence assistant (Feedly Leo) to search the web for content on obesity, cancer, hypertension, and diabetes (see Section 2.3.2 for details).

#### 2.3.1. Features of the PREDHiCT Mobile Application

The PREDHiCT app (see Figure 2): (1) monitors important lifestyle and health behaviors (such as physical activity and sleep); (2) provides personalized tips on how to prevent or manage risk for chronic diseases such as cancer, heart disease, diabetes, or obesity; and (3) connects you with opportunities to fight chronic disease through research. The PREDHiCT App, which is available for iOS and Android, was developed by our team and TrialX, a research mHealth company. The app also has the following components: personalized messages, a clinical trial repository, and a personalized newsfeed on chronic health conditions to increase health literacy and knowledge. To ensure the validity and authenticity of health, wellness, and lifestyle content within the PREDHiCT app, we meticulously integrated evidence-based practices from reputable sources such as the American Heart Association, American Diabetes Association, and the American Academy of Sleep Medicine. By adhering to these established guidelines, we upheld the quality and reliability of the information presented, bolstering the app’s credibility as a trustworthy resource for users seeking accurate health guidance.

#### 2.3.2. Personalized Newsfeed and Push Notifications

The integration of personalized newsfeed and push notifications within the PREDHiCT mobile application marks a transformative approach to enhance user engagement and knowledge dissemination. Through the synergistic utilization of existing consumer tools, including Feedly and the TrialX research platform, we established a dynamic avenue for delivering personalized health messages and tailored newsfeeds to research participants. Employing AI-driven technology via Feedly’s assistance bot Leo, we meticulously curated content aligned with participants’ interests, keywords, and similar boards. Leo’s training, spanning from 2020 to 2021, involved meticulous scanning of articles on predetermined boards, fine-tuning its AI model to achieve a minimum accuracy and precision of 80% using keywords such as cancer, diabetes, health, hypertension, diet, exercise, nutrition, food, therapy, sleep apnea, and sleep. Each selected article not only met this high standard but also aligned with predetermined health conditions and keywords, ensuring its relevance and pertinence to participants’ specific health interests and needs. The resultant newsfeeds were generated through RSS feeds for each board, ensuring thematic congruence and relevance. This innovative method optimally delivers current, accurate, and engaging content to participants, enriching their experience within the clinical trial journey. 

#### 2.3.3. Validation of AI-Based Newsfeed

To validate the newsfeed, a three-step process was employed. Initially, keywords were selected to generate lists of relevant articles, requiring consensus from four out of five study staff members on the articles’ relevance, their coverage of health facts, preventive strategies, and readability at an 8th-grade level. Articles meeting these criteria were utilized to train the AI algorithm over a 13-week period to enhance accuracy and precision. Before deployment, each newsfeed exceeded a 70% accuracy threshold, and further refinement was achieved by imposing more stringent parameters. Within this project, exclusionary terms such as COVID-19 were introduced. AI newsfeeds underwent 13 weeks of training, observing accuracy plateauing as depicted in Figure 3: hypertension (68% to 94%), diabetes (63–91%), cancer (0–70%), and exercise and diet (49% to 81%). Current learning scores for the Leo A.I. assistance are as follows: cancer 90%, diabetes 93%, hypertension 85%, and exercise and diet 91%, as illustrated in Figure 3. Private RSS URL and the Feedly API were employed to present Feedly articles in the app, extracting article titles, publication dates, images (if available), and original article URLs.

### 2.4. Phase 3: Randomized Controlled Trial

Recruitment. A total sample of 100 participants (50 each in the control and intervention arms) were recruited over a period of 18 months during the heights of the COVID-19 pandemic. We recruited participants from Research Match, social media platforms (Facebook, Twitter, Instagram, LinkedIn), email outreach via community connections, patient registries, and community organizations. Initial phone screenings gauged eligibility criteria, which included age (18+), English proficiency, smartphone ownership, app download willingness, and a self-reported history of cardiometabolic risk factors. Cardiometabolic risk factors included the following: hypertension, CAD, heart attack, heart failure, stroke, atherosclerosis, arrhythmias, PAD, type 2 diabetes, metabolic syndrome, obesity, dyslipidemia, insulin resistance, or elevated cholesterol. Eligible participants received an email link for informed consent and baseline surveys through REDCAP. Afterward, randomization placed them in either the personalized intervention or standard control group. To support participant retention, regular phone and email interactions were maintained with less-engaged participants. 

Intervention and controlled arms. In the intervention arm, participants experienced a multifaceted approach: they were exposed to a tailored newsfeed containing insights, advice, and solutions related to chronic health conditions such as hypertension, diabetes, cancer, and obesity. This feed also provided information about health events and clinical trials through an Eventbrite API linked with our mobile application featuring AI-curated content from the web on cancer, diabetes, obesity, and hypertension. A notable feature of our app was its integration with Fitbit or Apple Watch, motivating participants to monitor their sleep and physical activity for enhanced daily well-being. This engagement strategy ensured sustained interest in health-related content, clinical trial opportunities, and educational activities. Participants further received personalized SMS push notifications guiding them to track physical activity levels throughout the project duration. A resident physician was available to address medical queries from participants concerning their health status within the context of the study. In the control arm, participants received a standard paper-based treatment and did not have access to the personalized newsfeed or mobile application features described above. At follow-up assessments, adherence to physical activity and sleep was collected using self-reported diaries. Weekly ecological momentary assessments through mobile device messages gauged attitudes towards clinical trials over four weeks. Baseline demographic data was collected, and quantitative measures were administered at the four-week follow-up (see Figure 4). Participants received gift cards worth up to $100 as incentives for study completion. 

In accordance with best practices for conducting and reporting randomized controlled trials (RCTs), we diligently adhered to the Consolidated Standards of Reporting Trials (CONSORT) guidelines tailored to our study, PREDHiCT. CONSORT, developed to facilitate clear, transparent, and comprehensive reporting of RCTs, comprises a 25-item checklist. Our adherence to the CONSORT checklist ensured meticulous attention to detail in reporting various aspects of our trial design, *a priori* power analysis, *a posteriori* analysis, and interpretation, thus enhancing the rigor and transparency of our research methodology.

#### 2.4.1. Measures

The Self-Report Altruism Scale. The Self-Report Altruism Scale is a 20-item survey. Participants rate their frequency of altruistic behaviors using the categories never, once, more than once, often, and very often. The measure has high internal consistency (Cronbach alpha = 0.89) [27].

Research Attitude Questionnaire. The Research Attitude Questionnaire is an 11-item survey, rated on a 5-point Likert scale (1 = strongly disagree to 5 = strongly agree), that assesses respondents support for and value of research (Cronbach alpha = 0.75) with higher scores representing a more favorable attitude toward research [28,29].

Clinical Trials Vignette. Participants were presented with three clinical trial vignettes on diet, hypertension, and cancer. Participants were asked to rate their willingness to participate in a hypothetical clinical research trial. Responses were dichotomized into categories of “unwilling to participate” (responses 1, 2, or 3) and “willing to participate” (responses 4 or 5). 

All Aspects Health Literacy Survey. The All Aspects Health Literacy Survey (AAHLS) is a 13-item health literacy tool developed to understand an individual’s understanding of basic health information. Items are measured on a 3-point Likert scale with responses ranging from “rarely” to “often”; “Yes, definitely” to “Not really”; and “yes” “no”. The measure has adequate reliability (Cronbach’s alpha = 0.74). The reliability of the subscales was inconsistent [30]. 

eHealth Literacy Scale. The eHealth Literacy Scale (e-HEALS) is an 8-item scale that assesses an individual’s ability to search, combine, evaluate, and use health information from electronic sources, including the internet [31]. The eHEALS survey items are rated on a 5-point Likert scale, with responses ranging from “Strongly agree” to “Strongly disagree” (Cronbach alpha = 0.88). 

Health Trials Survey. The Health Trials Survey is a 6-item measure that measures attitudes towards participation in a clinical trial. Respondents rated items using five response categories, from “Strongly Agree” to “Strongly disagree”.

Socio-demographic survey. We collected participant age, gender, race/ethnicity, marital status, education level, employment status, income, medical care, past year major stressors, and substance use.

#### 2.4.2. Timeline

We collected survey data (above measures) at baseline and one month after enrollment for the main study hypotheses. We also captured attitudes about clinical trials weekly for 4 weeks through an ecological momentary assessment sent via message on the subject’s mobile device. This is to ascertain when the participant’s views changed about clinical trials over the course of the study. 

#### 2.4.3. The Impact of the COVID-19 Pandemic on Trial

Eligible subjects for all phases of the study consented before enrollment. Since Phase 1 of the study occurred before the COVID-19 pandemic, all focus groups were held in person. All data were de-identified and stored on a secure, password-protected drive, with access only available to the research study team. However, due to the COVID-19 pandemic and the need to maintain social distance, all procedures in Phase 3 occurred virtually or telephonically. 

#### 2.4.4. Statistical Analysis

We conducted various analyses to calculate the mean values along with standard deviations for continuous variables, as well as percentages for categorical variables. Differences in demographic characteristics of participants in control and intervention groups were compared by a two-samples *t*-test for continuous variables and a chi-squared test for binary variables, respectively. To evaluate the pre- and post-measurement differences between the control and intervention groups, a two-samples *t*-test was implemented. A *p*-value smaller than 0.05 was considered statistically significant. All statistical analysis was performed using R Statistical Software (version 3.6.1). 

## 3. Results

### 3.1. Qualitative Study Results for Phases 1 and 2

Barriers and Facilitators. The qualitative study’s thematic analysis provided a deep understanding of barriers and facilitators that shape participants’ decisions regarding clinical trial involvement (see Figure 2). Six latent themes were elucidated: (1) use of mobile phones; (2) utility and utilization of health and wellness digital applications (3) knowledge and experience being part of traditional clinical trials; (4) beliefs and attitudes on receiving digital notifications on mobile device; (5) the use of and value of health and wellness websites; and (6) recommended features and tools in clinical trial mobile applications (see Figure 5). These insights shed light on the potential of decentralized trials and mobile applications for effectively addressing these challenges. Mobile phone usage emerged as a notable theme, revealing that participants commonly use their devices for various purposes, including staying informed and accessing information. This indicates that a well-designed mobile app can leverage participants’ familiarity with their devices to deliver tailored health-related content and trial information directly to them, potentially increasing engagement.

The theme surrounding knowledge and experience with traditional clinical trials revealed barriers arising from limited awareness and uncertainties. A decentralized trial’s advantage lies in its potential to provide easily accessible and comprehensive educational materials tailored to participants’ conditions, thereby overcoming knowledge gaps and dispelling uncertainties. Moreover, the mixed attitudes towards digital notifications highlighted the importance of balance—the mobile app could capitalize on push notifications as reminders for trial-related activities while being mindful of participants’ preference for non-intrusive alerts.

The value participants placed on health and wellness websites underscored the potential for a mobile app to become a reliable hub for trial information and health-related resources. By centralizing trustworthy content and addressing concerns regarding information credibility, the app can effectively overcome a common barrier tied to information source difficulties. Participants’ recommendations for personalized features and a holistic approach within the app align with the decentralized trial’s potential to offer tailored experiences, addressing both facilitators and barriers. In essence, the qualitative findings emphasize how a decentralized trial, coupled with a well-designed mobile app, can collectively create an ecosystem that dismantles barriers, amplifies facilitators, and caters to participants’ preferences. This approach not only aligns with the changing landscape of health-related information consumption but also reflects a participant-centric strategy that has the potential to revolutionize clinical trial engagement.

### 3.2. Randomized Controlled Trial Results

The demographic characteristics of all participants were summarized in Table 2, which do not show any inherent differences between the control and intervention arms. The demographic characteristics showed no significant difference between the control and intervention groups. The average age of the sample was 53 years old, predominantly female (66.7%), White (66.7%), had at least a high school diploma or above, currently employed, and had an annual family income greater than $60,000. About half of the sample reported being married, and 26% were unmarried. 

Table 3 indicates that participants in the intervention arm had greater altruism and clinical trial literacy scores at the end of the study as compared to those in the control arm. 

Table 4 reports the pre- and post-measurement differences for the control and intervention groups. The pre- and post-change in the Self-Report Altruism Scale (mean: −2.94 vs. 0.83; *p*-value = 0.011), and clinical trial literacy (mean: 0.55 vs. 2.59; *p*-value = 0.001) were both significantly different between the control and intervention groups. In fact, altruism increased over the course of the study for participants in the intervention arm but significantly decreased for the control arm. While the pre- and post-change in clinical trial vignette and willingness were not significantly different between the control and intervention groups. 

## 4. Discussion

The current remote decentralized trial utilized a mixed-methods approach to develop a mobile application that provided novel education, navigation, and access experiences about clinical trials to a sample of 98 participants who have a personal or family history of a cardiometabolic condition. One of the primary innovative features of the intervention’s mobile application, PREDHiCT, is that it uses an artificial intelligence bot to curate a vast amount of content daily from the internet to provide personalized and tailored educational materials to participants about clinical trials and a variety of health conditions. A second unique feature of PREDHiCT is its intimate navigation of participants throughout the clinical trial journey. PREDHiCT weaves traditional health behavior tracking (sleep and physical activity), general health education about health conditions they care about, and call-to-action opportunities to combat health conditions the participant cares about—a strategy that engenders altruistic feelings about and willingness to participate in a trial. Ultimately, participating in a clinical trial is linked to health promotion behavior. Doing so highlights a third unique feature of PREDHiCT, which is making clinical trial information and opportunities more accessible to participants. Our findings indicate that these three unique features increased altruism and clinical trial literacy, but not willingness to participate in a trial. 

### 4.1. Literacy and Clinical Trials

Awareness and knowledge about clinical trials are considered primary drivers of clinical trial participation. Our study presents innovative approaches for disseminating educational materials and enhancing clinical trial literacy. These approaches differ from previous strategies, which typically adopt a broad, text-heavy, and jargon-laden format, limiting accessibility to specific learning styles and modes of reasoning, lacking engagement, and failing to offer interactive experiences for individuals to explore and understand clinical trials. We contend that enhancing clinical trial literacy is an evolving process that requires the utilization of emerging technologies like AI-powered personalized newsfeeds. These newsfeeds continually adapt to individuals’ preferences, adjusting factors such as message timing, frequency, and media type (e.g., animation or web articles) to ensure exposure to content that resonates with them. Findings from our current study indicate that participants in the PREDHiCT intervention arm had higher levels of clinical trial literacy and greater improvement in literacy compared to the control group at the end of the study. In fact, exposing individuals to pamphlets about clinical trials, as was the case in the control group, lowered literacy levels by the end of the study. Although it is unclear what contributed to this decrease, we conjecture that it is likely that participants in the control group were seeking inaccurate information about clinical trials and thus may have a flawed view of clinical trials. This highlights the need to have a well-curated and trusted platform to disseminate information about clinical trials. 

### 4.2. Altruism and Clinical Trials

Altruism is considered be another important driver of clinical trial participation. Altruism is an ethical value that participants use to justify their participation. People who participate in clinical trials do so because they want to help others with a similar condition through research. This hypothesis was demonstrated in several cancer, dementia, and chronic disease trials where a majority of participants expressed that altruism was the primary driver for participant enrollment in a clinical trial [32]. The impact of altruism on clinical trial participation occurs independent of sociodemographic, psychosocial, and clinical features. For our current study, the PREDHiCT solution significantly improved altruism compared to the control group. The observed changes in levels of altruism in our study debunk the idea that altruism is solely an innate virtue. In fact, our study provides innovative strategies as to how we can modify or engender altruistic feelings about clinical trial participation. Although we did not test which components of our PREDHiCT solution impacted changes in altruism, we believe our curated and tailored content about opportunities to fight specific chronic diseases and learning how each person can promote health and wellness for themselves and their network played significant roles. 

### 4.3. Advantages of a Remote and Virtual Decentralized Trial

Although we did not test the effectiveness of a remote and virtual decentralized trial experience on willingness to participate in a clinical trial, it is likely that it impacted our findings. The increased use of technology and digital health tools, including the use of websites and mobile apps, may provide a solution to address the ongoing lack of diversity in clinical research, thereby bridging the communication gaps and providing useful information to improve cardiometabolic outcomes among racial/ethnic minorities. Mobile technologies are scalable and pioneering approaches to reducing the burden of cardiometabolic diseases. However, evidence regarding their effectiveness is limited. A recent systematic review found that tracking daily health behaviors, disease education, self-monitoring and managing symptoms, and managing communication between providers are appealing app features. Thus, understanding the facilitators of optimizing digital self-care tools via mobile platforms is central to widening appeal, utility, and relevance [33]. A 2017 Research America national poll survey found that 53% of African-Americans, 50% Asians, and 58% Hispanics, compared to 39% of non-Hispanic whites, preferred to receive clinical trial information delivered to their mobile devices. The overwhelming evidence of the utility of virtual decentralized spaces to recruit participants supports our decision to conduct a decentralized trial that is remote and virtual.

### 4.4. Limitations

Findings from our study should be viewed with caution in light of several methodological limitations. First, while our use of remote decentralized controlled trials (RDCT) is seen as a positive attempt to be more inclusive of historically underrepresented groups in clinical trials, it can have some unintended consequences. For example, RDCTs may transfer the trial activity burden onto participants and remote-working research staff; therefore, additional support may be needed [21]. Second, it is likely that the presence of COVID-19, a global pandemic that captured the attention of the world, primed participants to see the value of medical research and thus biased their views about trials. Future studies could investigate the impact COVID-19 has on attitudes about clinical trials. Third, our study sample was small and thus not generalizable to a national population. Therefore, it is likely that our findings may only be applicable to individuals who already have a positive view of clinical trials, since a significant number of them were recruited through virtual channels like Research Match and by individuals in their social networks. Notwithstanding these limitations, we strongly believe our findings fill an important gap in the literature, whereby we now know that decentralizing the clinical trial process and providing tailored and dynamic educational, navigation, and access experiences significantly improved clinical trial literacy and altruistic views about trials. 

## 5. Conclusions

In conclusion, our study developed and tested the effectiveness of a digital app called PREDHiCT that aimed to improve clinical trial literacy and psychological predictors of clinical trial participation. We conducted focus groups to identify barriers and facilitators to clinical trial participation and developed a mobile application that contained innovative methods to educate, navigate, and increase access to clinical trial information and opportunities. Our decentralized trial involving 98 participants revealed that the PREDHiCT app significantly increased clinical trial literacy and altruism towards clinical trial participation. However, it did not significantly increase the willingness to participate in a clinical trial. These findings suggest that tailored education, navigation, and access to clinical trials, which are unique features of our PREDHiCT app, can enhance altruism and clinical trial literacy.

## Figures and Tables

**Figure 1 ijerph-20-07115-f001:**
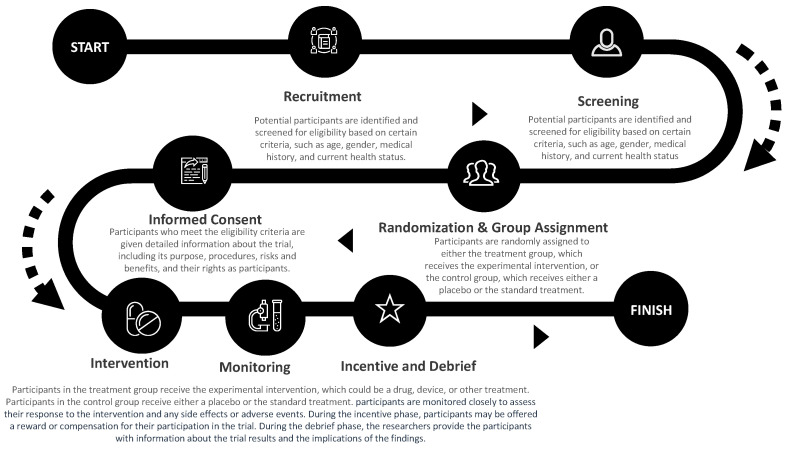
Plain-language journey roadmap distributed to participants to identify barriers and facilitators to clinical trial participation.

**Figure 2 ijerph-20-07115-f002:**
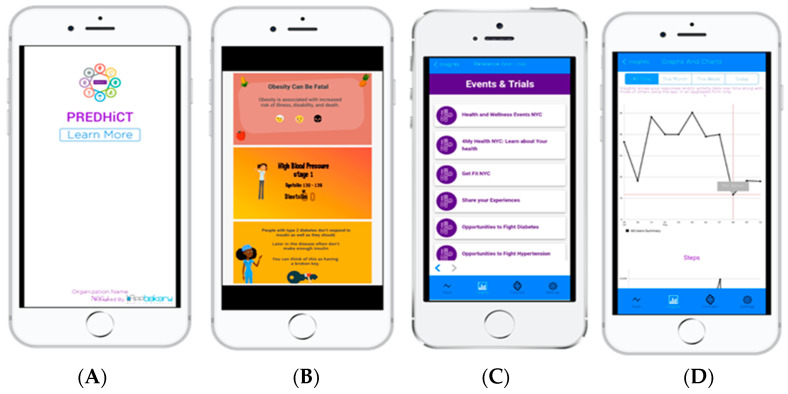
(**A**) PREDHiCT app homepage; (**B**) educational materials on chronic diseases, (**C**) events and clinical trial opportunities; (**D**) sleep and activity trends.

**Figure 3 ijerph-20-07115-f003:**
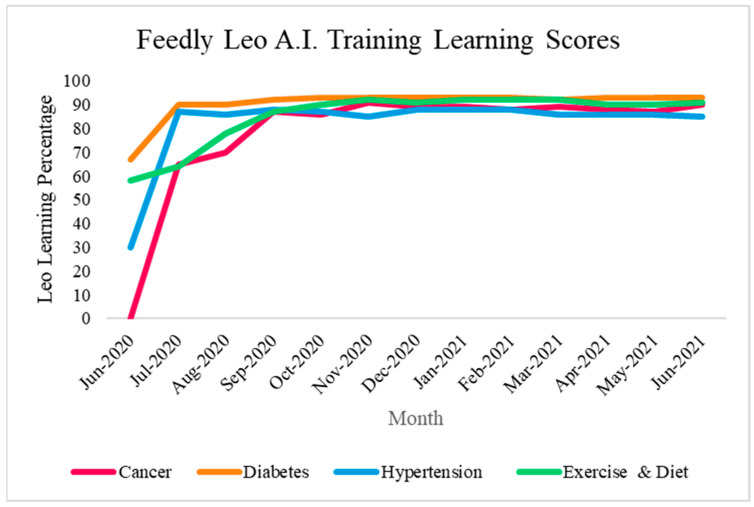
Accuracy of an AI-based newsfeed over time.

**Figure 4 ijerph-20-07115-f004:**
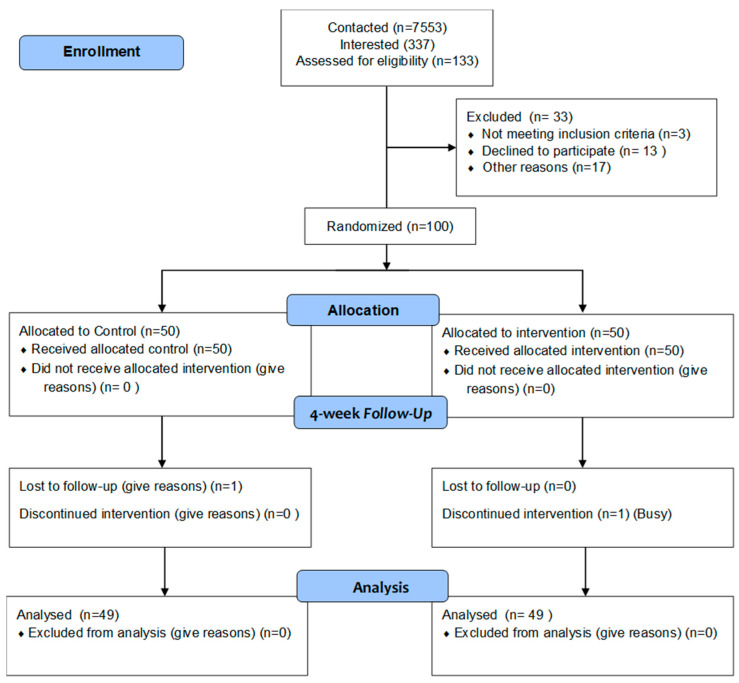
CONSORT flow diagram for PREDHiCT study.

**Figure 5 ijerph-20-07115-f005:**
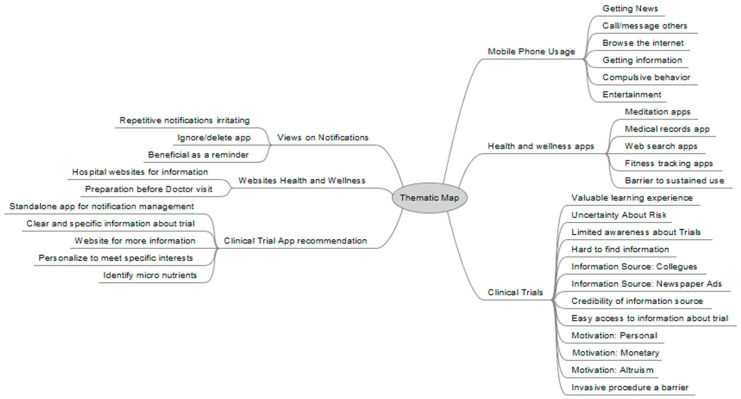
Thematic map of barriers and facilitators of decentralized clinical trials.

**Table 1 ijerph-20-07115-t001:** Phase 1 Focus group questions.

Domains	Questions
Mobile Phone Usage	How often do you use your phone? To check the news? For what purpose other than social media? To check your notifications? On average, how many hours a day do you spend on your phone? How often do you download and use apps on your smartphone?
Health and Wellness Apps/Websites	Do you regularly use health and wellness apps and websites to keep track of your lifestyle (sleep, exercise, etc.)? If so, what types of apps or websites do you use and at what frequency? What are some of the reasons you use apps? How helpful/not helpful have they been?
Views on Notification	How do you feel about receiving notifications? How many notifications would you prefer to receive a day?
Clinical Trials	Have you heard about clinical trials? ○ What was the source: doctor/media/internet/relatives/friends/colleagues/other: ○ Do you know what a clinical trial is? Yes/No ○ Have you ever participated in a clinical trial? Yes/No ○ Do you know anyone who participated in a clinical trial? ■ If so, how many? What do you think is the purpose/benefit of participating in a clinical trial? What would be your motivation/incentive (besides being paid) for participating in a clinical trial?
Clinical Trials App Recommendation	What are your expectations of an app where you can receive information about clinical research? What should it be like? What information would you like to see/receive to make a decision to participate in a clinical trial? What types of media would you like to receive when learning about clinical trials? In what ways can you engage in research/scientific activities? How do you find studies to participate in (word-of-mouth, television, radio, online, social media, bulletin boards, etc.?)

**Table 2 ijerph-20-07115-t002:** Demographic characteristics of participants.

	Overall	Control	Intervention	*p*-Value
	N = 98	N = 49	N = 49	
Age (mean (SD))	53.05 (16.02)	54.84 (16.33)	51.07 (15.64)	0.278
Gender: male (%)	31 (33.3)	19 (39.6)	12 (26.7)	0.271
Education: high school and above (%)	97 (97.0)	48 (96.0)	49 (98.0)	1.000
Employed (%)	63 (64.3)	33 (67.3)	30 (61.2)	0.673
Income above $60,000 (%)	65 (66.3)	33 (67.3)	32 (65.3)	1.000
Marital status (%)				0.550
Married	53 (53.0)	27 (54.0)	26 (52.0)	
Widowed	2 (2.0)	0 (0.0)	2 (4.0)	
Divorced	13 (13.0)	8 (16.0)	5 (10.0)	
Separated	3 (3.0)	2 (4.0)	1 (2.0)	
Unmarried	26 (26.0)	11 (22.0)	15 (30.0)	
Race (%)				0.562
African American/Black	30 (30.6)	15 (30.6)	15 (30.6)	
White	57 (58.2)	28 (57.1)	29 (59.2)	
Asian	2 (2.0)	2 (4.1)	0 (0.0)	
Non-white Hispanic	2 (2.0)	0 (0.0)	2 (4.1)	
Other	7 (7.1)	4 (8.2)	3 (6.1)	

Note: 100 participants were enrolled, and 2 participants dropped out; thus, the final sample size is 98.

**Table 3 ijerph-20-07115-t003:** Post-measurement score between the control and intervention groups.

	Control	Intervention	*p*-Value
Altruism sum (mean (SD))	40.78 (11.52)	45.82 (10.98)	0.042 *
Clinical trial vignette (mean (SD))	233.74 (66.40)	236.69 (57.32)	0.854
Willingness (mean (SD))	4.19 (1.12)	4.38 (0.99)	0.398
Literacy (mean (SD))	10.04 (2.77)	12.13 (1.67)	<0.001 *

* represent statistically significant finding.

**Table 4 ijerph-20-07115-t004:** Change in scores between control and intervention groups.

	Control	Intervention	*p*-Value
Altruism sum (mean (SD))	−2.94 (6.88)	0.83 (6.52)	0.011 *
Clinical trial vignette (mean (SD))	0.52 (53.49)	−3.59 (48.67)	0.757
Willingness (mean (SD))	−0.11 (1.17)	0.13 (1.10)	0.342
Literacy (mean (SD))	0.55 (2.67)	2.59 (2.41)	<0.001 *

* represent statistically significant finding.

## Data Availability

The data presented in this study are available on request from the corresponding author. The data are not publicly available due to strict ethical review board policies which requires a data use agreement before data are shared with anyone who was not part of the approved study.

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
