# Peer review of "Precision Recruitment and Engagement of Individuals at Risk for Diabetes and Hypertension in Clinical Trials (PREDHICT): A Randomized Trial for an E-Persuasive Mobile Application to Inform Decision Making about Clinical Trials"

_ijerph, 2023, doi:10.3390/ijerph20237115_

Round 1

Reviewer 1 Report (Previous Reviewer 2)

Comments and Suggestions for Authors

The authors have made some changes and improved the manuscript. However, the methods are still very confusing. Part 2.2.4 does not belong to the Methods section because it presents the results of Phase 1. It should be moved to the appendix section, but Figure 2 can stay in the Methods section with a short explanation in the text. There must be a way to shorten the Introduction (4 and a half pages!) and Methods section (9 pages!!!) because now it looks like a thesis, not a manuscript for some scientific journal. I like this study, but the authors should put more effort into making this study ready for publishing in this journal. Try making more tables and/or figures that could replace this huge text.  

Author Response

Reviewer 1

The authors have made some changes and improved the manuscript. However, the methods are still very confusing. Part 2.2.4 does not belong to the Methods section because it presents the results of Phase 1. It should be moved to the appendix section, but Figure 2 can stay in the Methods section with a short explanation in the text. There must be a way to shorten the Introduction (4 and a half pages!) and Methods section (9 pages!!!) because now it looks like a thesis, not a manuscript for some scientific journal. I like this study, but the authors should put more effort into making this study ready for publishing in this journal. Try making more tables and/or figures that could replace this huge text. 

Response: We thank reviewer 1 and significantly reduced the Introduction and re-arranged the Methods section.

Reviewer 2 Report (Previous Reviewer 3)

Comments and Suggestions for Authors

line 34 - Requires correction : "medical. Surgical or behavioral"

First paragraph should be shorter. There is a lot of unnecessary explaining.

I do not understand how this app can help the diversity in clinical trials. This sentence needs further elaboration.

The article (especially introduction) is too long. It needs to be shortened. Only most important points should be emphasized and hence included. There is a lot of repetition throughout the article in one way or the other which makes it difficult to read.

The article seemed to be general overview of approach to clinical trials until the line 156. In my opinion, this can be mentioned earlier. 

line 247 - How many focus groups there were?

In Materials and methods again seems to be a lot of repetition (Phase I). A lot of information was mentioned before. 

Does the artificial intelligence assistant search the web for medically relevant information or does it include whatever article or web site mentions these diseases?

It seems like the app leaves a little time and space to be dedicated to clinical trials. I do understand the need for additional content since people do not understand the benefits of clinical trial involvement but all of this seems too much. 

line 309 - "Use of mobile phone use."

Phase 1 Results of Focus Group - Includes good content but still needs sum up. 

Comments on the Quality of English Language

Minor English revision needed.

Author Response

Reviewer 2

line 34 - Requires correction : "medical. Surgical or behavioral"

First paragraph should be shorter. There is a lot of unnecessary explaining.

Response: Thanks to Reviewer 2, we made all the changes.

I do not understand how this app can help the diversity in clinical trials. This sentence needs further elaboration.

Response: We provided more details in Section 1.3.

The article (especially introduction) is too long. It needs to be shortened. Only most important points should be emphasized and hence included. There is a lot of repetition throughout the article in one way or the other which makes it difficult to read.

The article seemed to be general overview of approach to clinical trials until the line 156. In my opinion, this can be mentioned earlier.

Response: We made the changes to summarize. Thank you.

line 247 - How many focus groups there were?

Response: One focus group was conducted over two sessions. Please see in Section 2.2.1

In Materials and methods again seems to be a lot of repetition (Phase I). A lot of information was mentioned before.

Response: We made sure we removed redundant language.

Does the artificial intelligence assistant search the web for medically relevant information or does it include whatever article or web site mentions these diseases?

Response: Please see section 2.3.2 for detailed description of the AI-assistant.

It seems like the app leaves a little time and space to be dedicated to clinical trials. I do understand the need for additional content since people do not understand the benefits of clinical trial involvement but all of this seems too much.

Response: I appreciate the reviewers concerns but participants did not express this concern.

line 309 - "Use of mobile phone use."

Response: Thank you for this. We moved this to the Results section for the Qualitative study.

Phase 1 Results of Focus Group - Includes good content but still needs sum up.

Response: We thank the reviewer for this feedback. We believed that we provided sufficient description and summary of the results and to provide additional information would make the manuscript longer.

Reviewer 3 Report (New Reviewer)

Comments and Suggestions for Authors

The authors developed and tested the effectiveness of PREDHiCT app to  improve  clinical  trial  literacy  and  psychological  predictors  of clinical trial participation. The research is interesting and meaningful. But there are concerns.

1. The average age of the participants is over 50 years old, how to ensure that they can use the app correctly?

2. Medical problems have a certain degree of professionalism. How can patients effectively understand the designed problems and make correct choices? Is there any professional guidance?

3.How to ensure the validity and authenticity of the information provided by the subject?

4.What benefits can subjects get from the app? Do subjects have access to professional medical guidance? How to maintain subject compliance and long-term follow-up?

Author Response

Reviewer 3

The authors developed and tested the effectiveness of PREDHiCT app to  improve  clinical  trial  literacy  and  psychological  predictors  of clinical trial participation. The research is interesting and meaningful. But there are concerns.

  1. The average age of the participants is over 50 years old, how to ensure that they can use the app correctly?

Response: Yes we provided a training to use app and had FAQs and made ourselves available if people needed help with app or any technical aspects of the application.

  1. Medical problems have a certain degree of professionalism. How can patients effectively understand the designed problems and make correct choices? Is there any professional guidance?

Response: The study had a study physician and if participants had any medical questions the study physician answered these. I provided language in the Intervention Section 2.4.

3.How to ensure the validity and authenticity of the information provided by the subject?

Response: We added information in section 2.3.1

‘To ensure the validity and authenticity of health, wellness, and lifestyle content within the PREDHiCT app, we meticulously integrated evidence-based practices from reputable sources such as the American Heart Association, American Diabetes Association, and American Academy of Sleep Medicine. By adhering to these established guidelines, we upheld the quality and reliability of the information presented, bolstering the app's credibility as a trustworthy resource for users seeking accurate health guidance.”

4.What benefits can subjects get from the app? Do subjects have access to professional medical guidance? How to maintain subject compliance and long-term follow-up?

Response: Subjects using the app can gain several benefits, including access to evidence-based health, wellness, and lifestyle information derived from reputable sources such as the American Heart Association, American Diabetes Association, and American Academy of Sleep Medicine. The app empowers users to make informed decisions about their health and engage in behaviors that promote overall well-being. While the app offers valuable information and guidance, it is important to note that subjects do not have direct access to professional medical guidance through the app. We have a study physician if needs be. Instead, the app serves as an informative resource that complements medical advice from healthcare professionals.

To ensure subject compliance and facilitate long-term follow-up, the app employs various strategies. These strategies include personalized digital nudges that provide timely reminders and prompts to engage with health-related content and track behaviors. The incorporation of gamification elements, such as progress tracking and goal setting, enhances user engagement and motivation. Additionally, the app's user-friendly interface and diverse multimedia content contribute to sustaining subject interest over time. All of the above is described in the manuscript.

This manuscript is a resubmission of an earlier submission. The following is a list of the peer review reports and author responses from that submission.

Round 1

Reviewer 1 Report

Comments and Suggestions for Authors

The manuscript described the development and application of a mobile application to conduct a randomized human clinical trial.

Line 214 to 229 is somewhat problematic. The authors did not perform a thorough check before final submission to the journal.

None of the content is related to diabetes and hypertension.

The proportion of methodology to results is problematic. Results are poorly presented. 

References are limited, and they were not properly formatted in the text.

Figure 4 is presented not according to the Consort 2010 statements. 

Limited discussion. 

Reviewer 2 Report

Comments and Suggestions for Authors

Although it was done on a very small sample, the manuscript is in principle an interesting and important indicator of how well-designed mobile applications can influence the change of people's perception of clinical trials. However, there are a few minor things that need to be corrected before accepting the work:

a) References are not written in accordance with the journal's instructions! They should also be corrected numerically in the text according to the instructions.

b) I believe that the text written under 1.2 to 1.5 does not belong to the introduction, but should be reformulated and written in the methods, because they explain the ideas and reasons related to conducting the research and the author's opinion.

c) Lines 215-229 refer to general instructions to authors and have nothing to do with the written work. I believe this ended up in the manuscript by mistake.

d) The entire description of the methods is too long. A good part of the methods can be published as an appendix.

The paper is written clearly, critically clarifying its limitations. After these minor changes, the paper can be accepted for publication. 

Reviewer 3 Report

Comments and Suggestions for Authors

Language revision is required.

In Abstract there is no need for pointing out headlines (objective, method...). 

line 39-47: I do not see how these facts clarify the purpose of the article.

The app plan seems pretty wide, almost unfocused. Health and wellness application with tracking sleep and physical activity patterns is already used idea. Also, by choosing participants who would be more prone to trial enlistment because they are more physically active seems like narrowing the sample which was mentioned before is not something to strive to. 

The general idea seems like it has got potential but the data are all over the place. Please clarify the primary objectives and make it simple.

Comments on the Quality of English Language

Needs extensive language revision.
